# Enhanced Intervertebral Disc Repair via Genetically Engineered Mesenchymal Stem Cells with Tetracycline Regulatory System

**DOI:** 10.3390/ijms242216024

**Published:** 2023-11-07

**Authors:** Yeji Kim, Seong Bae An, Sang-Hyuk Lee, Jong Joo Lee, Sung Bum Kim, Jae-Cheul Ahn, Dong-Youn Hwang, Inbo Han

**Affiliations:** 1Research Competency Milestones Program of School of Medicine, CHA University School of Medicine, Seongnam-si 13496, Republic of Korea; ygknow.k@gmail.com; 2Department of Neurosurgery, CHA University School of Medicine, CHA Bundang Medical Center, Seongnam-si 13496, Republic of Korea; hanib@chamc.co.kr; 3Department of Biomedical Science, Graduate School of CHA University, Seongnam-si 13496, Republic of Korea; humit159@naver.com; 4Department of Medicine, Graduate School, Kyung Hee University, Seoul 02453, Republic of Korea; kkeith123@naver.com; 5Department of Neurosurgery, Kangbuk Samsung Hospital, Sungkyunkwan University College of Medicine, Seoul 03181, Republic of Korea; 6Department of Neurosurgery, Kyung Hee University, Seoul 02453, Republic of Korea; sungbumi7@hanmail.net; 7Department of Otorhinolaryngology-Head and Neck Surgery, CHA University School of Medicine, CHA Bundang Medical Center, Seongnam-si 13496, Republic of Korea; 8Department of Microbiology, School of Medicine, CHA University, Seongnam-si 13496, Republic of Korea

**Keywords:** mesenchymal stem cells, tetracyclin (Tet) regulatory system, Tet-off, intervertebral disc regeneration

## Abstract

The therapeutic potential of Mesenchymal stem cells (MSCs) for the treatment of Intervertebral disc (IVD) degeneration can be enhanced by amplifying specific cytokines and proteins. This study aimed to investigate the therapeutic potential of tetracycline-off system-engineered tonsil-derived mesenchymal stem cells (ToMSC-Tetoff-TGFβ1-IGF1-BMP7) for treating intervertebral disc (IVD) degeneration. ToMSCs were isolated from a tonsillectomy patient and genetically modified with four distinct plasmids via CRISPR/Cas9-mediated knock-in gene editing. Transgene expression was confirmed through immunofluorescence, western blots, and an enzyme-linked immunosorbent assay for transforming growth factor beta 1 (TGFβ1) protein secretion, and the effect of MSC-TetOff-TGFβ1-IGF1-BMP7 on disc injury was assessed in a rat model. The ToMSC-Tetoff-TGFβ1-IGF1-BMP7 treatment exhibited superior therapeutic effects compared to ToMSC-TGFβ1, and ToMSC-SDF1α implantation groups, stimulating the regeneration of nucleus pulposus (NP) cells crucial for IVD. The treatment showed potential to restore the structural integrity of the extracellular matrix (ECM) by upregulating key molecules such as aggrecan and type II collagen. It also exhibited anti-inflammatory properties and reduced pain-inducing neuropeptides. ToMSC-Tetoff-TGFβ1-IGF1-BMP7 holds promise as a novel treatment for IVD degeneration. It appears to promote NP cell regeneration, restore ECM structure, suppress inflammation, and reduce pain. However, more research and clinical trials are required to confirm its therapeutic potential.

## 1. Introduction

Intervertebral disc (IVD) degeneration is a prevalent condition often associated with chronic lower back pain (LBP) that meaningfully impacts a considerable proportion of the adult population [1]. This ailment imposes a substantial socioeconomic burden on both individuals and society as a whole [2]. IVD degeneration can be managed with surgical or non-surgical methods, but current treatments focus on temporary symptom relief rather than addressing the underlying cause [3,4]. Persistent or recurrent LBP after current treatments can lead to complications and continued IVD degeneration [4,5]. Therefore, ongoing research is exploring potential therapies to address the underlying causes of IVD degeneration.

The etiology of IVD degeneration is complex, involving factors such as genetic predisposition, biochemical aspects, and mechanical stressors, including spinal injuries or trauma [6]. IVDs, situated between the vertebrae, are crucial for enabling spinal movement and flexibility. The IVD structure consists of the nucleus pulposus (NP) surrounded by the annulus fibrosus (AF) and bordered by cartilaginous endplates. This configuration maintains a delicate balance between anabolic and catabolic activities, regulating the organization of extracellular matrix (ECM) components, which include type II collagen and proteoglycans [7]. However, this balance is disrupted in cases of IVD degeneration, triggering the progressive deterioration of ECM components and structural alterations within the IVD [8,9]. The loss of ECM components undermines spinal flexibility and elasticity, resulting in the formation of clefts and tears within the AF [10]. Consequently, this causes the protrusion and herniation of the NP, ultimately leading to the symptoms of LBP and radiating leg pain.

IVD degeneration is typically managed through surgical procedures or non-surgical strategies, such as physical therapy and medication. The chosen method of treatment depends on various factors, including the severity of LBP, the presence of radiating leg pain, radiological findings, the patient’s overall health status, and individualized considerations [9]. However, current treatments primarily aim to provide temporary relief of LBP and leg pain symptoms rather than targeting the underlying pathogenic processes involved in IVD degeneration. Surgical options include procedures such as discectomy and spinal fusion. Discectomy, with or without spinal fusion, is the most commonly performed surgical treatment for lumbar disc herniation and lumbar stenosis [3,4]. Despite the use of these surgical treatments, persistent or recurrent LBP and/or leg pain can occur due to the development of post-spinal surgery syndrome (PSSS). This condition is reported in 10–40% of patients undergoing spinal procedures, with the incidence varying based on the specific procedure and individual factors [11]. The complexity of the surgery generally impacts the prevalence of PSSS, with discectomy showing an incidence rate of 19–25% [12]. Furthermore, 10–30% of patients who undergo discectomy experience subsequent IVD degeneration and associated LBP [5]. The reported incidence of PSSS following spinal fusion ranges from 30% to 46% [13]. As a result, spinal surgery can cause increased spinal instability and additional pressure on adjacent discs, leading to further IVD degeneration [4]. Current treatments, therefore, do not effectively restore the structure and characteristics of the degenerated disc to a non-degenerated state, highlighting an important unmet need in addressing the root causes of IVD degeneration. At present, various potential treatments such as growth factors, gene therapy, stem cells with or without biomaterials, and senolytics are being explored for IVD degeneration [14]. Mesenchymal stem cells (MSCs) are adult stem cells derived from a variety of sources that can differentiate into adipocytes, osteoblasts, and chondrocytes [15]. MSCs are regarded as immune privileged cells because they do not express major histocompatibility complex-II (MHC-II) and costimulatory molecules [16]. Previous studies have demonstrated their anti-inflammatory, regenerative, and immunomodulatory potential [17,18,19]. Therefore, MSC therapy is a potentially promising treatment for IVD degeneration.

Reactive oxygen species act as crucial mediators in cell-to-cell signaling within degenerated discs, affecting various cellular processes such as matrix metabolism, apoptosis, the pro-inflammatory phenotype, and disc cell senescence. An overproduction of reactive oxygen species leads to oxidative stress, a key contributor to IVD degeneration [20]. This oxidative stress triggers inflammation and matrix degeneration and modulates matrix proteins, resulting in oxidative damage to the disc’s ECM and mechanical impairments within the IVD [21]. Consequently, the excessive oxidative stress-driven activation of inflammatory factors accelerates IVD degeneration [22]. Therefore, an urgent need exists for research focusing on treatments that promote anti-inflammatory and immunomodulatory functions to counteract IVD degeneration [23]. Mesenchymal stem cells (MSCs) present a promising research focus. MSCs are adult stem cells derived from various tissues, including bone marrow, adipose tissue, and the umbilical cord [24]. The widespread accessibility and scalability of MSCs have contributed to their active exploration in research [25]. According to recent research, the transplantation of MSCs into rat models has been shown to delay the progression of IVD degeneration [26]. Furthermore, it was demonstrated that the regulation of the expression of specific transcription factors can accelerate chondrogenesis in an IVD degeneration rat model [27]. Notably, several preclinical studies have recently assessed the therapeutic potential of MSCs in treating IVD degeneration using animal models [28]. MSCs have the capacity to migrate to damaged IVD tissue and exhibit potent anti-inflammatory and immunomodulatory properties [19]. Furthermore, they can differentiate into chondrogenic cell types, stimulating high rates of proteoglycan production and adopting characteristics of NP-like cells [29,30]. As MSCs differentiate into a chondrogenic cell type, chondrogenic markers such as aggrecan and type II collagen, which are major components of the ECM in the IVD regions, are highly expressed [27]. Therefore, the differentiation of a chondrogenic cell type through MSCs treatment can accelerate the regeneration of cartilage tissue and the synthesis of matrix in degenerated IVD. MSCs also play a role in preserving and proliferating NP cells within degenerated IVDs by upregulating NP marker gene expression and promoting ECM synthesis [31]. Through their paracrine effect, MSCs effectively inhibit apoptosis and foster regeneration and repair of damaged IVD tissue. Consequently, MSC therapy has emerged as a promising approach for treating IVD degeneration. However, when MSCs are transplanted alone into degenerated IVD, MSCs have difficulty to survive because of the low oxygen, low pH, low glucose levels, and excessive inflammatory mediators at the degenerated IVD environment [32].

The therapeutic potential of MSCs for the treatment of IVD degeneration may be enhanced by amplifying specific cytokines and proteins. If repeated injections of specific cytokines or proteins are used for increasing the therapeutic potential of MSCs, they can trigger another inflammatory response and further injury [33]. Thus, it is important to develop genetically engineered MSCs with specific cytokines or proteins to regulate expression of specific factors and increase the therapeutic effectiveness of MSCs. Previous research has employed a technique involving the transplantation of retroviral tetracycline (Tet)-dependent regulatory systems, such as the Tet-off vector, into bone marrow-derived MSCs (BMSCs) to modulate the expression of stromal cell-derived factor 1 (SDF-1) [34]. The Tet-off system involves a tetracycline-controlled transactivator (tTA) that binds to the tetracycline response element (TRE) promoter in the absence of tetracycline. Tetracycline repressor protein (TetR) represses the expression of the tetA gene and regulates the expression of the tetracycline resistance genes, by binding to the tetA operator. The TetR portion of the Tet-off transcription factor binds very specifically to its target sequence and does not activate off-target cellular genes. This high specificity may be due in part to the prokaryotic nature of these components, which act in the context of large eukaryotic genomes without similar elements. In the absence of doxycycline, the Tet-off system turns off the TRE promoter. Consequently, a specific gene inserted into the vector is expressed in the absence of tetracycline, while its expression is inhibited upon tetracycline treatment. For well-selected clones, the maximum induction of the Tet-off system is several thousand-fold and can be detected within a few hours after removal of doxycycline from the culture medium [35]. In contrast to effectors used in other systems, such as ecdysone, doxycycline is inexpensive, well-characterized, and produces highly reproducible results. Doxycycline binds with high affinity to Tet-off system elements and is essentially non-toxic at effective concentrations. This temporal gene regulation approach mitigates the risk of continuous gene expression and related physiological side effects, which is a common concern with other genome editing technologies such as the clustered regularly interspaced short palindromic repeats (CRISPR)/CRISPR-associated protein 9 (Cas9) system [36]. To briefly summarize, the system uses a gene that is not expressed in the presence of doxycycline but is expressed in the absence of doxycycline. The Tet-off system allows the regulation of gene expression levels at specific time points through the administration of tetracycline or its analogs, such as doxycycline, thereby mitigating potential physiological side effects.

In this study, our objective was to improve the therapeutic efficacy for IVD degeneration by incorporating Tet-off expression vectors into Tonsil-derived mesenchymal stem cells (ToMSCs) to modulate the expression of key factors. Tonsil-derived MSCs have been proposed as an excellent source of MSCs for clinical applications due to the fact that cells can be obtained in smaller amounts compared to bone marrow-derived stem cells (BMSCs) and Adipose-derived stem cells (ADSCs) [37]. The proliferation rate is relatively high compared to BMSCs and ADSCs, and the immunogenicity is lower compared to the previous two cells [38]. The osteogenic, adipogenic, and chondrogenic abilities of ToMSCs have also been validated in numerous studies [39,40,41]. For those reasons, the ToMSCs were used in this study. These factors, including transforming growth factor beta 1 (TGF-β1), insulin-like growth factor 1 (IGF-1), and bone morphogenetic protein-7 (BMP-7), have been demonstrated to elicit anti-inflammatory and growth-promoting effects within the IVD. TGF-β1 promotes NP cell proliferation and ECM synthesis while suppressing the expression of a disintegrin and metalloproteinase with thrombospondin motifs (ADAMTS) -4 and -5, enzymes involved in aggrecan degradation [42]. IGF-1 stimulates the synthesis of ECM and proteoglycans [43]. Additionally, BMP-7 enhances aggrecan and type II collagen gene expression and promotes glycosaminoglycan synthesis in NP cells [44]. Current research suggests that among the many growth factors known to have a regulatory effect on chondrogenesis, TGF-β proteins are the most potent inducers of chondrogenesis in hMSCs. In contrast, other growth factors appear to mediate chondrocytic physiology rather than promote chondrogenesis in human mesenchymal stem cells (hMSCs) [45,46,47,48,49]. We decided to identify factors that could regulate the main mechanism, TGFβ/BMP signaling. TGF-β1 is one of the strongest candidates, and we selected IGF-1 and BMP-7 as candidates together, which could have a synergistic effect on chondrogenesis [50,51]. We chose these three elements because they are the most used key elements in cartilage differentiation, and we expect them to play a much larger role in cartilage differentiation in vivo and in vitro than if there were no genes or only one gene. Considering these roles, we proposed that MSCs equipped with the Tet-off system could effectively counteract ECM degradation in IVD degeneration, promote disc tissue regeneration, and simultaneously inhibit the secretion of inflammatory factors, thereby providing pain relief.

## 2. Results

### 2.1. Immunophenotypic Characterization of ToMSCs

ToMSCs were prepared from tonsil tissue and characterized using flow cytometric analysis. These ToMSCs showed typical spindle-shaped MSC morphology (Figure 1A) and were set up for flow cytometry to exclude dead cells and debris (Appendix A). As negative controls, we used appropriate isotype controls for each antibody (Appendix A), which were positive for MSC surface markers including CD44, CD73, CD90, and CD105. Consistent with the characterization of MSCs, ToMSCs did not express HLA class II (DRB1) but did express HLA class I (HLA-ABC) antigens (Figure 1B). CD44, CD73, CD90, and CD105, which were positive for MSC surface markers, were analyzed by flow cytometry statistics (Appendix A).

### 2.2. CRISPR/Cas9-Mediated Knock-In of Transgenes into a Safe-Harbor Site on the ToMSC Chromosome

To insert the gene, we used AAVS1, which is the most used and verified safe-harbor site that does not affect other chromosomes [52]. Previous studies have shown that ToMSCs exhibit chondrogenic potential under suitable conditions [39,53]. In the present study, we aimed to enhance these capabilities by genetically modifying ToMSCs. We overexpressed several factors that may be involved in stem cell homing and chondrogenesis, including SDF1α and TGFβ1 under the CAG promoter along with a fusion cDNA TGFβ1-P2A-IGF1-T2A-BMP7 under the Tet-off promoter. These modifications were integrated into the adeno-associated virus integration site 1 (AAVS1) locus of the ToMSC chromosome, a safe-harbor site, using CRISPR/Cas9-mediated homology-directed repair (Figure 2A). When the gene cassette was correctly inserted, the genomic polymerase chain reaction (PCR) was performed in such a way that the PCR band could be confirmed at 1.1 kb (Sequences of PCR primers and sgRNA are in Appendix A). If the gene is not inserted, the band would not appear like the negative control, ToMSC and D.W (Figure 2B).

### 2.3. Assessment of Transgenic Expression through qRT-PCR, Western Blots, Immunocytochemistry, and ELISA

The successful expression of inserted transgenes in ToMSCs was confirmed using western blots (Figure 3A), immunocytochemistry (Figure 3B), and qRT-PCR (Figure 3C). In ToMSC-SDF1α-6H and ToMSC-TGFβ1-6H cells, SDF1α-6H and TGFβ1-6H proteins were, respectively, detected by an antibody against the 6His tag through western blots and immunocytochemistry (Figure 3A,B). In ToMSC-Tetoff-TGFβ1-IGF1-BMP7 cells, BMP7 expression was observed only in the absence of doxycycline (Figure 3A,B). qRT-PCR was performed to measure the relative expression of TGFβ1, IGF1, and BMP7 among ToMSC, ToMSC-Tetoff-TGFβ1-IGF1-BMP7 (In the absence of doxycycline), ToMSC-SDF1α, and ToMSC-TGFβ1 (Figure 3C). In the absence of doxycycline, TGFβ1, IGF1, and BMP7 transgenes were overexpressed from the ToMSC-Tetoff-TGFβ1-IGF1-BMP7 cell line, when compared to ToMSC (Figure 3C) (ToMSC vs. Tet off ToMSC TGFB1: 1.0 ± 0.04 vs. 1.83 ± 0.09), (ToMSC vs. Tet off ToMSC IGF1: 1.0 ± 0.05 vs. 1.50 ± 0.08), (ToMSC vs. Tet off ToMSC BMP7: 1.0 ± 0.06 vs. 1.35 ± 0.05).

ELISA analysis revealed that ToMSC-TGFβ1 cells expressed TGFβ1 at a level twice as high as naïve ToMSCs (484 ± 15 vs. 243 ± 3) (Figure 4). The ToMSC-Tetoff-TGFβ1-IGF1-BMP7 cells exhibited a 1.5-fold higher expression of TGFβ1 compared to naïve ToMSCs when doxycycline was omitted from the medium (375.5 ± 12.5 vs. 243 ± 3) (Figure 4).

### 2.4. ToMSC-Tetoff-TGFβ1-IGF1-BMP7 Exhibited the Best Anti-Allodynic Effect

Two weeks after needle puncture disc injury at the Co6/7 and Co7/8 levels in the rat model, the Co7/8 discs in each group were injected with ToMSC-TGFβ1, ToMSC-SDF1α, or ToMSC-Tetoff-TGFβ1-IGF1-BMP7 (ToMSC-Tetoff). The Co6/7 discs were used as injury controls, while the normal controls consisted of Co5/6 discs. To evaluate the mechanical allodynia induced by the needle puncture disc injury, a von Frey test involving a filament was conducted on the ventral surface of the tail. In previous studies, the significant behavioral changes related to pain were observed from one week after inflicting traumatic injury on the IVD or performing treatment on degenerated discs [54,55,56]. In this study, mechanical allodynia was evaluated using the von Frey test at 7-day intervals following the implementation until scarifying. This test was performed 1 day prior to injection (D-1) and at 7-day intervals after implantation, up to 41 days. IVD degeneration is known to lead to discogenic LBP and radicular leg pain [20]. As degeneration progresses, pain sensitivity increases and pain-related behaviors emerge [57]. The 50% withdrawal threshold refers to the mechanical force required for a response in which 50% of the experimental animals withdraw their paws. A lower 50% withdrawal threshold value (g) indicates that the animals are more sensitive to pain and can experience mechanical allodynia in degenerated IVD. Conversely, a higher 50% withdrawal threshold signifies that the pain sensitivity has been alleviated due to the implantation at the IVD degeneration model [58]. In the present study, no significant differences were detected among the treatment groups on the day prior to implantation (D-1) up to day 28 post-implantation. However, at 35 and 41 days post-implantation, the 50% withdrawal threshold was markedly lower in the ToMSC-TGFβ1 and ToMSC-SDF1α implantation groups (Figure 5), suggesting a significant increase in pain sensitivity associated with all treatments except for ToMSC-Tetoff implantation.

### 2.5. ToMSC-Tetoff-TGFβ1-IGF1-BMP7 Demonstrated the Best Restoration of Disc Anatomy and Hydration in the Rat Tail Needle Puncture Injury Model

Six weeks after implantation, in vivo T2-weighted MRI was performed to assess the regeneration of degenerated discs. The MRI index is defined as the NP area multiplied by the average signal intensity. To quantify degeneration in the NP region, the MRI index is employed [56,59]. A higher T2-weighted MRI index value indicates that the NP area is well-preserved and appears to be in a more normal condition. The MRI index was calculated for the normal control disc (Co5/6), the injured control disc (Co6/7), and the Co7/8 disc of each group that had received treatment materials. As observed on axial and coronal T2-weighted MRI, the MRI index of the ToMSC-Tetoff group was significantly higher compared to the injury-only, ToMSC-TGFβ1, and ToMSC-SDF1α groups (Figure 6). From the axial perspective, the ToMSC-Tetoff implantation group exhibited higher signal intensity in the NP regions compared to the other groups. The ToMSC-Tetoff treatment group demonstrated the most effective disc anatomy restoration and regeneration, as seen on T2-weighted MRI (Figure 6A,B). Moreover, the ToMSC-Tetoff treatment group displayed superior disc hydration, suggesting that the water content in the ToMSC-Tetoff treatment group surpassed that of the other treatment groups.

Histological analysis was performed on rat coccygeal discs. H&E staining was employed to evaluate the NP, AF, and overall morphology of the IVD, while Safranin O staining was used to assess the proteoglycan distribution in the NP region. Healthy disc tissue exhibited a well-defined boundary created by the AF tissue, which enclosed normal NP cells, as demonstrated by H&E staining. In contrast, H&E staining of injury-only disc tissue showed a reduction in NP cell count due to IVD degeneration. Furthermore, NP and AF were indistinguishable from one another (Figure 6C). The ToMSC-Tetoff treatment group, in comparison to the other implantation groups, displayed a higher NP cell count and formed a relatively distinct boundary with the AF tissue. Upon evaluating histological scores based on H&E and Safranin O staining results [60], the ToMSC-Tetoff implantation group exhibited significantly lower histologic scores compared to the injury-only, ToMSC-TGFβ1, and ToMSC-SDF1α implantation groups (Figure 6D). This group also maintained a statistically significant larger NP area compared to the injury-only and ToMSC-TGFβ1 treatment groups (Figure 6E). Safranin O staining revealed a high proteoglycan content in the NP area of healthy disc tissue. In contrast, the injury-only disc tissue demonstrated a considerable decrease in proteoglycan content. Among the implanted groups, the ToMSC-Tetoff group showed superior proteoglycan content in the NP region (Figure 6C).

These findings suggest that ToMSC-Tetoff treatment may play a role in both delaying and repairing degenerated disc tissue, as well as preventing the breakdown of matrix components, including proteoglycans. According to the previous research, structural changes in the IVD disrupt the nutritional supply to disc tissue and contribute to the acceleration of the IVD degeneration [61]. Additionally, structural changes lead to abnormal growth of blood vessels and neural tissue within the IVD [62]. Consequently, structurally degenerated IVD is more prone to experiencing mechanical allodynia [63]. ToMSC-Tetoff treatment enhances the regeneration of the NP tissue within the IVD, preserving matrix hydration and attenuating anatomical changes. This result suggests that the ToMSC-Tetoff treatment attenuates the degeneration of the IVD, abnormal growth of nerve tissues, and the occurrence of mechanical allodynia. As a result, ToMSC-Tetoff therapy could potentially be a promising approach for the repair of human intervertebral disc tissue.

### 2.6. ToMSC-Tetoff-TGFβ1-IGF1-BMP7 Demonstrated the Best Preservation of Matrix Proteins in the Disc NP of the Rat Tail Needle Puncture Injury Model

Aggrecan and type II collagen, two critical components of the ECM, are essential for maintaining water retention in disc tissue and preserving the resilience and volume of the NPs. During IVD degeneration, inflammation-related factors such as IL-1β, IL-6, and TNF-α trigger ECM remodeling, leading to a reduction in aggrecan and type II collagen content [64]. In the present study, the immunohistochemical staining results revealed a statistically significant reduction in aggrecan and type II collagen content in the injury-only disc tissue compared to the healthy disc tissue (Figure 7A). When comparing the implantation groups, the ToMSC-Tetoff group displayed a significantly higher content of type II collagen compared to the injury-only and ToMSC-TGFβ1-treated disc groups. Notably, the ToMSC-Tetoff group exhibited the highest median aggrecan and type II collagen-positive area among the implantation groups (Figure 7B,C). These findings suggest that ToMSC-Tetoff treatment has the potential to restore the matrix composition of IVDs, contributing to the maintenance of NP volume and the recovery of elasticity.

### 2.7. ToMSC-Tetoff-TGFβ1-IGF1-BMP7 Exhibited the Best Preservation of Endogenous Disc NP Progenitor Cells

Immunofluorescence staining for Brachyury and Tie2 was conducted on each disc tissue to assess the presence of NP cells displaying endogenous phenotypes. Brachyury is a crucial transcription factor involved in notochord formation during embryonic mesoderm development and remains highly expressed in NP cells even after NP cell formation is complete. Moreover, Brachyury is known to regulate the expression of factors associated with IVD degeneration [65]. Tie2, possessing progenitor-like multipotency, promotes the expression of aggrecan and type II collagen, and protects NP cells from apoptosis, thus contributing to the maintenance and survival of NP cells [66,67]. Staining results demonstrated a statistically significant higher expression of Brachyury and Tie2 in healthy disc tissue compared to the injury-only disc tissue (Figure 8A). Among the implantation groups, the ToMSC-Tetoff group exhibited significantly higher expression of Brachyury and Tie2 than the injury-only and ToMSC-TGFβ1 implantation disc groups (Figure 8B,C). These findings suggest that ToMSC-Tetoff treatment enhances Brachyury and Tie2 expression and could potentially contribute to the formation and maintenance of NP cell survival.

### 2.8. ToMSC-Tetoff-TGFβ1-IGF1-BMP7 Displays the Greatest Cell Proliferation

Immunofluorescence staining for human nuclear antigen was performed, allowing for an assessment of the retention of the implanted cells. This enabled the evaluation of the proliferation of NP cells and the formation of ECM tissues from ToMSCs [68]. The staining results revealed that the ToMSC-Tetoff treatment group exhibited a statistically significant increase in human nuclear antigen expression compared to the other implantation groups. In contrast, no human nuclear antigen expression was detected in the healthy disc and injury-only disc tissues (Figure 9A,B). The positive cell ratio of mCherry in the ToMSC-mCherry treatment group was barely expressed. This could be attributed to decreased cell proliferation and viability in the acidic pH environment of the degenerative IVD [69]. The very low expression of mCherry did not affect the results of the red fluorescence staining in this study.

In previous research, MSCs do not elicit a strong inflammatory response in the allo or xenotransplantation. Thus, MSCs are considered hypoimmunogenic [70]. In the rat model, xenotransplantation of the human MSCs promoted the remyelination of spared white matter, the enhancing axonal growth in spinal cord, and the regeneration of calvarial bone [71,72]. This indicates that even when human MSCs are xenotransplanted into a rat model, the human MSCs are capable of surviving and differentiating. This suggests that the xenotransplantation of human MSCs into a rat needle puncture injury disc model was successfully achieved, and implanted human MSCs remain viable and promote the proliferation of NP cells and ECM structures, even 6 weeks after ToMSC-Tetoff implantation into the rat IVD.

### 2.9. ToMSC-Tetoff-TGFβ1-IGF1-BMP7 Demonstrated the Greatest Decrease in Catabolic Enzymes and Pro-Inflammatory Cytokines in the Disc NP

MMP-13 is a catabolic gene that plays a crucial pathological role in IVD degeneration, acting as an important factor in ECM catabolism. Abnormal expression of MMPs can disrupt the balance of the ECM, leading to a decrease in type II collagen [73]. Specifically, abnormal expression of MMP-13 in IVD degeneration accelerates the degradation of ECM components, including aggrecan and type II collagen [74]. In the present study, immunofluorescent staining revealed the lowest expression of MMP-13 in healthy disc tissue (Figure 10A). In contrast, injury-only disc tissue displayed the highest MMP-13 expression, constituting a statistically significant difference from both the implantation and healthy disc tissues. This was followed by the ToMSC-TGFβ1 and ToMSC-SDF1α treatment groups. Notably, the ToMSC-Tetoff treatment group exhibited a statistically significant reduction in MMP-13 expression compared to the other implantation groups (Figure 10B). These results support the hypothesis that ToMSC-Tetoff implantation can downregulate ECM catabolism and consequently attenuate the progression of IVD degeneration.

Immunofluorescence staining was performed for pro-inflammatory cytokines TNF-α and IL-1β to assess their expression in an IVD degeneration model. TNF-α and IL-1β serve as potent stimulators of pro-inflammatory factor release, promoting ECM degradation and IVD phenotype degeneration [75]. In degenerated IVDs, high TNF-α and IL-1β expression has been observed, and it has been suggested that early inhibition of TNF-α expression could alleviate the progression of IVD degeneration and associated pain [76]. Immunofluorescent staining results revealed that the ToMSC-Tetoff treatment group demonstrated statistically significantly lower TNF-α and IL-1β expression compared to the injury-only disc tissue (Figure 10C). Furthermore, when comparing the ToMSC-Tetoff treatment to other implantation groups, TNF-α exhibited statistically significantly lower expression than the ToMSC-TGFβ1 group (Figure 10D,E). These findings suggest that ToMSC-Tetoff treatment has the potential to downregulate the expression of inflammatory signaling.

According to previous research, MMP-13 is actively transcribed and activated in human NP cells in degenerative IVD. This leads to the secretion of pro-inflammatory cytokines such as IL-1β, IL-6, and TNF-α, as well as matrix-degrading enzymes [77]. In this study, the ToMSC-Tetoff treatment group exhibited lower expression of MMP-13 compared to other treatment groups. Thus, similarly to previous research findings, it was observed that the expression of pro-inflammatory cytokines, such as IL-1β and TNF-α, was reduced.

Accumulated pro-inflammatory cytokines weaken the microenvironment of the IVD by causing damage to the ECM. In previous studies, excessive expression of IL-1β was shown to alter the anatomical structure of the IVD and induce the mechanical allodynia [54]. Additionally, the expression of TNF-α accelerates cellular aging within the IVD and induces processes like apoptosis. The accumulated inflammatory stresses further stimulate the secretion of molecules like MMP-13, contributing to ECM degradation and worsening the IVD microenvironment [78].

In essence, the increased expression of MMP-13, TNF-α, and IL-1β accelerates cellular aging within the IVD and leads to the development of discogenic pain [77]. In this study, the ToMSC-Tetoff treatment suppressed the expression of MMP-13, TNF-α, and IL-1β compared to other treatment groups. This suggests that ToMSC-Tetoff treatment may clinically mitigate the structural degeneration of the IVD, inhibit the increasing in pain sensitivity, and reduce the occurrence of LBP.

### 2.10. ToMSC-Tetoff-TGFβ1-IGF1-BMP7 Shows the Greatest Reduction of Injury-Induced Pain

Immunofluorescence staining for CGRP was performed. During IVD degeneration, there is an increase in the expression of neuropeptides, such as CGRP, within the nociceptive pathway, which is associated with discogenic pain. As IVD degeneration progresses, the expression of CGRP and its receptor increases in disc tissue. In vitro experiments have shown that CGRP inhibits NP cell proliferation and induces apoptosis, as well as activating signaling pathways that promote inflammation and degenerative disc phenotypes [79]. In the present study, immunofluorescent staining revealed that CGRP expression was the lowest in healthy disc tissue, increasing in injured disc tissue (Figure 11A). Furthermore, the ToMSC-Tetoff treatment group exhibited significantly lower CGRP expression compared to the other treatment groups and injury-only disc tissue (Figure 11B). This suggests that ToMSC-Tetoff treatment may reduce CGRP expression, potentially decreasing chronic discogenic LBP and inflammation resulting from IVD injury.

## 3. Discussion

This study was conducted to explore the effects of regenerating disc tissue, suppressing inflammatory factors, and mitigating pain by implanting ToMSCs equipped with the Tet-off system into the NP area of discs affected by IVD degeneration. The results showed that the implanted MSCs in the ToMSC-Tetoff-TGFβ1-IGF1-BMP7 group remained viable for 6 weeks post-implantation, as indicated by high human nuclear antigen expression. These implanted MSCs, regulated by the Tet-off system, induced the expression of TGFβ1, IGF1, and BMP, which in turn promoted the proliferation of NP cells and ECM structures. Furthermore, these MSCs mediated the downregulation of inflammation-related factors and the expression of neuropeptides implicated in discogenic pain. Therefore, it was demonstrated that ToMSC-Tetoff-TGFβ1-IGF1-BMP7 implanted into the NP region of degenerated discs can not only regenerate disc tissue but also simultaneously exert anti-inflammatory and anti-allodynic effects.

In previous studies, the regeneration of degenerated discs has been explored through the administration of various factors, including growth factors. For instance, in a study by Wang et al., the introduction of TGFβ1 into cells harvested from patients undergoing spine surgery promoted NP cell proliferation and reduced the expression of ADAMTS-4 and -5 enzymes, which play a pivotal role in aggrecan degradation. In contrast, treating these harvested cells with IL-1β inhibited NP cell proliferation and significantly increased the expression of ADAMTS-4 and -5 enzymes, facilitating aggrecan degradation. Both TGFβ1 and IL-1β are involved in the synthesis and degradation of the ECM and have been suggested as potential therapeutic targets for the treatment of IVD degeneration [42]. In the current study, we employed the Tet-off system to stimulate the expression of TGFβ1 in MSCs. This led to enhanced NP cell proliferation and reduced ECM degradation compared to other treatment groups. The comparatively lower expression of IL-1β also supports the proposition that ToMSC-Tetoff treatment could delay the degradation of aggrecan, an ECM component. In another study conducted by Travascio et al., the exogenous administration of IGF-1 was proposed to treat IVD degeneration. IGF-1 is a notable anabolic agent that promotes proteoglycan production and IVD cell proliferation and plays a vital role in maintaining IVD homeostasis [80,81]. Applying IGF1 exogenously to pathologically degenerated discs under conditions of adequate nutrition has shown beneficial effects, including increased proteoglycan biosynthesis [43]. In our study, we induced the expression of IGF1 by employing a Tet-off system. In line with the previous study, our approach preserved NP cell proliferation and ECM components at a higher rate compared to the other treatment groups. In a study by Li et al., in vitro and in organ culture, the implantation of the BMP-2/7 heterodimer post-nucleotomy in bovine caudal discs stimulated gene expression and proteoglycan synthesis in the residual NP tissue [44]. In our study, BMP7 expression induced using the Tet-off system contributed to maintaining the volume and resilience of the NP area. When ToMSC-Tetoff-TGFβ1-IGF1-BMP7 was implanted in the degenerated rat coccygeal disc, the expression of each factor helped maintain the NP tissue structure, promote NP cell proliferation, and restore ECM components.

Several prior preclinical studies have explored the use of MSCs in animal models of IVD degeneration. For instance, Iimori et al. generated human-induced pluripotent stem cell (hiPSC)-derived cartilage and transplanted it into 3.5-mm defects created in the femurs of immunodeficient mice to evaluate its reparative capacity. The hiPSC-derived cartilage implanted in the mice sustained human nuclear antigen expression throughout the 28-week observation period and facilitated the formation of new bone to fill the bone defect [68]. Consequently, the expression of MSCs in the transplanted tissue is essential for the MSCs to fulfill their function. In the present study, transplanted MSCs remained viable within the IVD for up to 6 weeks post-implantation. This suggests that MSCs contributed to NP cell and ECM proliferation, as well as anti-inflammatory responses, during the observation period. In another study by Sun et al., the mechanism behind IVD degeneration improvement via MSC-derived extracellular vesicles (MSC-EVs) was examined. Incubation of MSC-EVs with TNF-α–intervened NP cells resulted in the downregulation of microRNA-194-5p (miR-194-5p) and the upregulation of tumor receptor-associated factor 6 (TRAF6). Restoring miR-194-5p enhanced proliferation and decreased apoptosis of TNF-α–intervened NP cells, whereas TRAF6 overexpression had a detrimental impact on the growth of TNF-α–intervened NP cells. Hence, protecting NP cells from TNF-α has therapeutic potential in the treatment of IVD degeneration [82]. Furthermore, in a study by Wang et al., IL-1β treatment of extracted cells inhibited NP cell proliferation and significantly increased the expression of ADAMTS-4 and -5 enzymes, promoting aggrecan degradation [42]. This suggests that the expression of TNF-α and IL-1β not only reduces aggrecan and type II collagen content but also triggers NP cell apoptosis [64]. In the present study, ToMSC-Tetoff implantation notably reduced TNF-α and IL-1β levels relative to injury-only disc tissue. In another study, Sun et al. assessed the effects of adipose-derived stromal cells (ASCs) on human NP cells. ASCs suppress the activation of MMPs (MMP-3 and MMP-13), disintegrin, and ADAMTSs (ADAMTS-1 and 5), along with pro-inflammatory factors (IL-1β, IL-6, TGFβ1, and TNF-α). Therefore, ASCs protect NP cells from compressive load and inhibit NP cell degeneration and death [83]. In a study by Le Maiter et al., the overexpression of MMP-13 accelerated the degradation of ECM components, including aggrecan and type II collagen, during IVD degeneration [74]. In this study, transplantation of MSCs utilizing the Tet-off system significantly inhibited the activation of MMP-13 and pro-inflammatory factors (IL-1β and TNF-α). This preserved NP cells from degeneration and contributed to the structural restoration of NP cells by inhibiting the degradation of aggrecan and type II collagen. According to a study by Sun et al., the expression of neuropeptides such as CGRP—implicated in the nociceptive pathway—increased during IVD degeneration, leading to discogenic pain [79]. Moreover, Takahashi et al. found that exposure of NPs to AF may induce neuronal damage, leading to hyperalgesic responses by promoting nerve growth factor-mediated axonal growth and inflammation [84]. Therefore, the reduction of CGRP-positive neurons is a crucial aspect of managing discogenic pain [85]. In our study, the ToMSC-Tetoff treatment group displayed the lowest CGRP expression compared to the other treatment groups. Additionally, the von Frey test indicated that the 50% withdrawal threshold at 35 and 41 days post-implantation was not significantly different from that of the healthy disc group. This suggests that ToMSC-Tetoff treatment significantly decreased pain sensitivity.

This study had several limitations. First, the needle puncture disc injury model in rat coccygeal discs does not fully replicate age-related disc degeneration. Accurately mimicking the multitude of factors involved, including various inflammatory mediators, low glucose levels, hyperosmolarity, and the low pH found in degenerated discs, is challenging [32]. These factors have the potential to inhibit the functional expression of transplanted MSCs. Second, the rat coccygeal disc differs significantly from the human lumbar disc, both mechanically and biologically [56]. Therefore, future research should employ primate animal models that more closely mirror the clinical course and mechanical properties of human age-related IVD degeneration. Third, further research is needed to understand the structure and inflammatory factor expression of the AF when damaged by needle puncture. In our study, we examined the restoration of NP cell composition and structure through MSC injection into the NP region. Although significant improvements were observed compared to injury-only tissue, the regeneration of NP cells alone may not be sufficient. If the partially damaged AF structure is not restored, increased NP proliferation and herniation can occur due to tears and clefts in the AF [86]. Fourth, it is essential to investigate the survival of transplanted MSC tissue over a longer experimental period [68]. Fifth, biomechanical tests evaluating the resilience of regenerated NP cells are necessary. Sixth, we performed fluorescence immunostaining to observe the expression of NP progenitor cells, human nuclear antigen, and proinflammatory cytokines. To support this, additional experiments such as ELISA, qPCR, and flow cytometry may be necessary. Seventh, the assessment of injury-induced pain through the analysis of the expression of various pain markers is necessary. In our study, we utilized CGRP. In the degenerated IVD animal model, CGRP is known to be overexpressed and capable of inducing LBP. Thus, CGRP is an assessment marker for injury-induced pain [87,88]. According to previous research, there is increased expression of inducible nitric oxide synthase (iNOS) and prostaglandin E2 in degenerated IVD [23]. Therefore, iNOS and prostaglandin E2 can be utilized as markers for evaluating LBP or injury-induced pain. Various experiments, including studies with longer experimental durations, will allow for a more comprehensive demonstration of the effectiveness of ToMSC-Tetoff treatment.

When applying ToMSC-TetOff treatment to humans, several specific issues are anticipated. The low oxygen, low pH, and diminished nutritional supplies within human degenerated discs are expected to impair the survival of MSCs within the human IVD [32,61]. Mechanical movements in the human lumbar region, such as weight-bearing, flexion, extension, and rotation, may hinder the survival and proper differentiation of injected MSCs in their intended locations. If the multiple MSC injections are required to overcome these issues, it must be considered that additional inflammatory responses and injuries may occur during the injection process [33]. To enhance the viability of MSCs within the human intervertebral disc (IVD) and induce stable differentiation, applying bio-scaffolds or a chemo-attractant delivery system can be further considered [89,90].

In the case of conducting long-term clinical trials in human subjects, the risk of the ectopic bone formation and microvascular embolism due to MSC differentiation within the human IVD must be considered [91]. Through our research, the tetracycline-off system is expected to mitigate these risks. First, ToMSC-Tetoff can be injected into the human IVD for inducing differentiation of disc tissue. After that, it is possible to suppress the expression of growth factors at a specific time point by injecting doxycycline to the same location. As shown in this study, ToMSCs without the growth factor expression cannot survive in the IVD for an extended period. Therefore, this approach can reduce the risk of ectopic bone formation and microvascular embolism associated with continuous MSC differentiation. To apply this approach to human clinical trials, further research is needed to determine the appropriate timing for the inhibition of growth factor expression.

Despite these limitations, this study remains unique in its application of MSCs using a tetracycline-off system for degenerated IVDs. In a normal system, it is necessary to consider how the above three factors will cause changes during cell growth, but by using the tetracycline-off system, we blocked the prevention of cell changes during cell growth and used this system to show a clear effect on chondrogenesis by expressing TGFB1-IGF1-BMP7 in the absence of tetracycline. In this study, MSCs with the tetracycline-off system were implanted into the NP region of degenerated IVD. ToMSC-Tetoff-TGFβ1-IGF1-BMP7 exhibited (1) high MSCs survival, (2) a reduction in secretion of catabolic enzyme and pro-inflammatory factors, and (3) a reduction in the expression of factors associated with the increasing pain sensitivity. Consequently, this research shows that ToMSC-Tetoff-TGFβ1-IGF1-BMP7 implantation maintenances the anatomical structure of IVD and suppressed chronic discogenic LBP, through NP cells regeneration and inhibition of NP tissue degeneration.

## 4. Materials and Methods

### 4.1. Isolation and Culture of Tonsil-Derived Mesenchymal Stem Cells

Tonsil-derived mesenchymal stem cells (ToMSCs) were isolated following a slightly modified version of a previously established protocol. Briefly, discarded tonsil fragments were obtained from children who had undergone tonsillectomy at the Department of Otolaryngology of Bundang CHA Hospital (Seongnam-si, Republic of Korea). Written informed consent was obtained from the parents of the children who provided tonsil tissue. The study protocol was approved by the institutional review board of Bundang CHA Hospital (approval number: 2021-03-016). Isolated tonsillar tissue was washed twice with 1 × phosphate-buffered saline (PBS) (Welgene, Gyeongsan-si, Republic of Korea). The tissue was then cut into small pieces and digested in RPMI 1640 medium (Invitrogen, Waltham, MA, USA) containing 210 U/mL collagenase type 1 (Invitrogen) and 10 μg/mL DNase I (Sigma-Aldrich, St. Louis, MO, USA) for 30 min at 37 °C. The digested tissue was filtered through a wire mesh, and the collected cells were washed twice with RPMI 1640/20% fetal bovine serum (FBS) (Gibco, Waltham, MA, USA), followed by an additional wash with RPMI 1640/10% FBS. Mononuclear cells were isolated from the cell suspension using Ficoll-Paque (GE Healthcare, Little Chalfont, UK) density gradient centrifugation, and 1 × 10^8^ cells were seeded in a T-125 culture flask containing Dulbecco modified Eagle medium F12 (Thermo Fisher Scientific, Waltham, MA, USA) supplemented with 10% FBS, 100 μg/mL streptomycin (Invitrogen), and 100 U/mL penicillin (Invitrogen). After 48 h, unattached cells were discarded, and the adherent cells were expanded with fresh medium.

### 4.2. Flow Cytometry Analysis

To characterize the phenotype of ToMSCs, flow cytometry analysis was performed. A minimum of 30,000 cells were suspended in 100 μL of 1 × PBS containing 2% FBS and stained with fluorochrome-conjugated antibodies. The antibodies used were APC-αCD44, PE-αCD73, PE-αCD90, APC-αCD105 (all from Miltenyi Biotec, Teterow, Germany), PE-αHLA-DR (BD Biosciences, Franklin Lakes, NJ, USA), and FITC-αHLA-ABC (Thermo Fisher Scientific).

### 4.3. Plasmid Construction

#### 4.3.1. pAAVS1-Puro-CAG-SDF1α-6H

To obtain the SDF1α-6H (6His) DNA, PCR was conducted using pBABE puro SDF-1 alpha (Addgene, Watertown, MA, USA) as a template, which produced a DNA fragment containing Sal I and Mlu I sites at the 5′- and 3′-ends, respectively. The PCR product was then cloned into the pTA vector (Real Biotech Corp, Banqiao City, Taiwan). Subsequently, the SDF1α-6H DNA fragment was isolated from the pTA-SDF1α-6H plasmid through Sal I/Mlu I double digestion. This fragment was inserted into a Sal I/Mlu I-digested pAAVS1-puro-CAG-EGFP plasmid (Addgene). The resulting plasmid, pAAVS1-puro-CAG-SDF1α, was prepared on a medium scale (midiprep) and confirmed by sequencing.

#### 4.3.2. pAAVS1-Puro-CAG-TGFβ1-6H

The pAAVS1-puro-CAG-TGFβ1-6H plasmid was generated using the same approach as the pAAVS1-puro-CAG-SDF1α-6H plasmid. In this instance, TGFβ1-6H complementary DNA (cDNA) was acquired through PCR, utilizing TGFB1_pLX307 (Addgene) as a template.

#### 4.3.3. pAAVS1-Puro-CAG-mCherry

The pAAVS1-puro-CAG-mCherry plasmid, used as a negative control, was generated using the same protocol as the pAAVS1-puro-CAG-SDF1α-6H and pAAVS1-puro-CAG-TGFβ1-6H plasmids. For this cloning process, mCherry DNA was PCR-amplified from pCDNA3-FlipGFP-Casp3-T2A-mCherry (Addgene).

#### 4.3.4. pAAVS1-Puro-Tetoff-TGFβ1-IGF1-BMP7-CAG-tTA-Advanced

The Tet-off promoter and tTA-Advanced gene were obtained from pTRE-TIGHT (Takara Bio, Shiga, Japan) and pAAV-Tetoffbidir-Alb-luc (Addgene), respectively. TGFβ1, IGF1, and BMP7 cDNAs were derived from TGFB1_pLX307 (Addgene), pT7T3Pac-IGF1 (Korea Human Gene Bank, Daejeon, Korea), and pTetON-BMP2/7 (Addgene), respectively. The three genes were fused using P2A and T2A sequences and expressed as a monocistronic gene cassette under the Tet-off promoter. Following this, all DNA components were subcloned into the pAAVS1-puro-CAG plasmid, resulting in the creation of pAAVS1-puro-Tetoff-TGFβ1-IGF1-BMP7-CAG-tTA-Advanced.

### 4.4. CRISPR/Cas9-Mediated Knock-In Gene Editing

The AAVS1 locus, a safe-harbor site, was utilized as the integration site for the knock-in gene cassettes using the CRISPR/Cas9 system [52,92]. Transfections were performed on ToMSCs (passage 2) with 6.25 μg of Cas9 protein, 12 μg of sgRNA, and 1 μg of donor DNA using the Neon Transfection System (Invitrogen) with parameters set at 990 V, 40 ms, and 2 pulses. The donor DNAs used in this study included pAAVS1-Puro-CAG-TGFβ1-6H, pAAVS1-puro-CAG-SDF1α-6H, and pAAVS1-puro-Tetoff-TGFβ1-IGF1-BMP7-CAG-tTA-Advanced. The resulting stable ToMSC lines were designated as ToMSC-TGFβ1-6H, ToMSC-SDF1α-6H, and ToMSC-Tetoff-TGFβ1-IGF1-BMP7, respectively.

Following electroporation, cells were treated with 0.7 μg/mL puromycin (Sigma-Aldrich) 48 h later to select for the knock-in colonies. The chosen clones were then cultured in the presence of 0.14 μg/mL puromycin.

### 4.5. Immunofluorescent Detection of Transgene Expression

Cells were washed with ice-cold 1 × PBS and fixed using 4% paraformaldehyde/1 × PBS (Biosolution, Seoul, Republic of Korea). The fixed cells were washed four times with 1 × PBS and subsequently permeabilized with 0.1% Triton-X (Sigma-Aldrich) for 10 min. Subsequently, the cells were blocked with 5% normal goat serum (Thermo Fisher Scientific) and incubated overnight at 4 °C with primary antibodies, under constant shaking. The primary antibodies used included anti-His tag antibody (1:2000) (ABM, Richmond, BC, Canada) for ToMSC-SDF1α-6H and ToMSC-TGFβ1-6H, as well as anti-BMP7 antibody (1:500) (Thermo Fisher Scientific) for ToMSC-Tetoff-TGFβ1-IGF1-BMP7. After incubation with the primary antibodies, cells were washed three times with 1 × PBS and incubated with a goat anti-rabbit Alexa Fluor 594 IgG antibody at room temperature for 1 h (1:200) (Invitrogen). Following four washes with 1 × PBS, the cells were counterstained with 4′,6-diamidino-2-phenylindole and dihydrochloride (DAPI) (1:2000) (Sigma-Aldrich) and observed under a microscope.

### 4.6. Western Blots

Total protein was extracted from cells using RIPA buffer (Thermo Fisher Scientific), and 30 µg of protein per lane was separated via 12% SDS-PAGE. The proteins were then transferred onto nitrocellulose membranes (Sigma-Aldrich). Subsequently, the membranes were blocked in 5% non-fat milk for 1 h at room temperature with constant shaking. This was followed by incubation with primary antibodies, including anti-His tag antibody (1:2000) (abm), anti-BMP7 antibody (1:500) (Thermo Fisher Scientific), and anti-β-actin (1:1000) (Santa Cruz Biotechnology, Dallas, TX, USA) at 4 °C overnight. The membranes were then washed three times with PBST (1 × PBS, 0.2% (*v*/*v*) Tween-20) and incubated with the corresponding secondary antibodies (goat anti-rabbit or goat anti-mouse IgG H&L (HRP) (LI-COR, Lincoln, NE, USA) at room temperature for 2 h. Afterward, the membranes were washed another three times with 1 × PBST, then finally analyzed using an Odyssey CLx Imager (LI-COR).

### 4.7. Quantitative Real-Time Polymerase Chain Reaction (qRT-PCR)

Total RNAs were extracted from samples of ToMSC-Tetoff-TGFβ1-IGF1-BMP7, ToMSC-TGFβ1, ToMSC-SDF1α, and ToMSC-mCherry using an RNA purification kit made by MACHEREY-NAGEL (MACHEREY-NAGEL GmbH & Co., Düren, Germany), and cDNAs were synthesized using 100 ng of each RNA using a cDNA synthesis kit (TOYOBO, Osaka, Japan). Quantitative RT-PCR was performed using a power SYBR green master mix (Thermo Fisher Scientific).

### 4.8. Enzyme-Linked Immunosorbent Assay

The level of TGFβ1 protein secreted into the medium from cultured cells was measured using a human TGFβ1 DuoSet enzyme-linked immunosorbent assay (ELISA) kit (R&D Systems, Minneapolis, MN, USA) according to the manufacturer’s instructions.

### 4.9. Animal Surgical Procedures

Thirty-five female Sprague-Dawley rats, 10 weeks old and weighing between 230 and 250 g, were purchased from Orient Bio. Inc. (Seongnam, Republic of Korea). The rats were acclimatized for 1 week on a 12/12 h light/dark cycle, with a temperature of 22 °C ± 1 °C and a relative humidity of 50% ± 1%. During this period, the animals were given free access to food and water.

The animal study was approved by the Institutional Animal Care and Use Committee of CHA University (IACUC220190) and was conducted in accordance with the committee’s guidelines.

Prior to surgery, the peritoneal site of the rat was sterilized with 70% alcohol, followed by the administration of general anesthesia via intraperitoneal injection of Zoletil (50 mg/kg; Virbac Laboratories, Carros, France) and Rompun (10 mg/kg; Bayer, Seoul, Republic of Korea). The tail skin was then sterilized with 70% alcohol and treated with povidone-iodine. A dorsal longitudinal skin incision was made along the tail to expose the coccygeal disc. The coccygeal discs (Co6/7, Co7/8) were punctured with a 21-G sterile spinal needle, from the dorsal to the ventral side of the tail, perpendicular to the tail skin. The needle was subsequently rotated 360° twice and held for 30 s [93,94,95]. The needle was bent at a length of 5 mm, limiting the puncture depth to that distance. The skin was then sutured with Nylon 4-0 and sterilized with povidone-iodine. Throughout the experiment, the rats were kept on healing pads to prevent hypothermia. Postoperatively, an analgesic (ketoprofen, 1 mg/kg; SCD Pharm. Co. Ltd., Seoul, Republic of Korea) and an antibiotic (Cefazolin, 15 mg/kg, CKD Pharmaceuticals, Seoul, Republic of Korea) were administered at appropriate doses for 3 days. All rats were housed individually under a 12/12 h light/dark cycle at a controlled temperature of 22 °C ± 1 °C and a relative humidity of 50% ± 1%.

### 4.10. Experimental Design in a Rat Disc Degeneration Model

Degeneration of the coccygeal disc (Co6/7 and Co7/8) was induced in 35 rats by puncturing the center of the disc using a 21-G spinal needle. The rats were then randomly divided into four groups: (1) ToMSC-mCherry (n = 7), (2) ToMSC-TGFβ1 (n = 9), (3) ToMSC-SDF1α (n = 9), and (4) ToMSC-Tetoff-TGFβ1-IGF1-BMP7 (n = 10). Two weeks after the induction of disc degeneration, ToMSC-mCherry, ToMSC-TGFβ1 (1 µL of 2 × 10^4^ cells + 1 µL hydrogel), ToMSC-SDF1α (1 µL of 2 × 10^4^ cells + 1 μL hydrogel), and ToMSC-Tetoff-TGFβ1-IGF1-BMP7 (1 µL of 2 × 10^4^ cells + 1 µL hydrogel) plasmids were implanted into the center of the punctured Co7/8 disc in each group using a Hamilton syringe with a 30-G needle. Six weeks post-implantation, the rats were euthanized via carbon dioxide asphyxiation, and the coccygeal discs were removed for radiologic and histologic analysis. The healthy IVDs at Co5/6 in each group were used as the normal control group, and the degeneration-induced Co6/7 discs in each group were used as the injury-control group.

### 4.11. Quantitative Behavioral Nociception Assay

The von Frey test for behavioral nociception was performed 1 day prior to implantation and then on days 7, 14, 21, 28, 35, and 42 post-implantation [58]. To minimize the potential misinterpretation of exploratory activities as a positive response, the rats were individually placed in small cages with mesh floors and lids to acclimate to the environment for 20 min. Subsequently, a 2-g filament was applied with sufficient force to the ventral surface of the tail for a maximum of 6 s. A response was considered positive if the rat exhibited nocifensive behaviors, such as flinching, licking, withdrawing, or shaking the base of the tail, immediately or within 6 s. In the event of a positive response, the rat was tested with a lighter filament. However, if no response was detected, testing continued with a heavier filament until five responses were recorded in five attempts for two consecutive filaments. The von Frey analysis was conducted by two independent observers who were blinded to the treatment conditions of the specimens. The average values of the specimens’ 50% withdrawal threshold detected in each treatment group were calculated. At the same time point ‘after implantation’, one-way analysis of variance (ANOVA) with the Tukey post-test was applied to the average values of each group.

### 4.12. Magnetic Resonance Imaging

Six weeks post-implantation, changes in disc structure and the degree of coccygeal disc degeneration were evaluated using a 9.4-T magnetic resonance imaging (MRI) spectrometer (Bruker BioSpec, Billerica, MA, USA). T2-weighted imaging was performed for the coronal plane with the following parameters: time to repetition (TR) of 5000 ms, time to echo (TE) of 30 ms, 150 horizontal × 150 vertical matrix, 15 horizontal × 15 vertical fields of view, and 0.5-mm slices with 0-mm spacing between each slice. For the sagittal plane, imaging was conducted with parameters including TR of 5000 ms, TE of 30 ms, 150 horizontal × 150 vertical matrix; 15 horizontal × 15 vertical field of view, and 0.5-mm slices with 0-mm spacing between each slice. The MRI index (calculated as the area of the NP multiplied by the average signal intensity) was used as an outcome measure to evaluate the extent of rat coccygeal disc degeneration [56,59]. The high signal intensity area in the mid-sagittal plane of the T2-weighted image, which delineates the NP, was considered the region of interest. The signal intensity of this region was measured using ImageJ (version 1.50b, National Institutes of Health, Bethesda, MD, USA; https://imagej.nih.gov/ij/) and compared to that of a normal rat coccygeal disc. The MRI index of each group was assessed by two independent observers who were blinded to the implantation in the rat coccygeal discs (n = 7 for group 1, n = 9 for group 2, n = 9 for group 3, and n = 10 for group 4).

### 4.13. Histological Analysis

At 6 weeks post-implantation, the rats were euthanized using excessive carbon dioxide inhalation, and coccygeal disc tissues, along with the adjacent vertebral body, were preserved in 10% neutral buffered formalin for 1 week. Following this, the tissues underwent decalcification in Rapid Cal Immuno (BBC Biochemical, Mount Vernon, WA, USA) for 2 weeks. The disc tissues were then embedded in paraffin and sectioned to a thickness of 6 µm using a microtome (Leica, Wetzlar, Germany). Next, the sections were dewaxed, rehydrated, and stained with Safranin O (Sigma-Aldrich) to evaluate the distribution and quantity of proteoglycan. Finally, the sections were mounted using mounting media and scanned with a C-mount camera adapter (U-TVO.63XC; Olympus, Tokyo, Japan).

Histological scoring was performed using a comprehensive 16-point scale to assess the IVD based on Safranin O staining [56,96]. This scale was divided into five subcategories, focusing on NP morphology, NP cellularity, AF morphology, endplate morphology, and the boundary between the NP and AF. Degenerative features in NP and AF morphology were given high importance, as were changes in notochordal cell morphology in degenerative rat IVDs; thus, these categories were weighted twice. Detailed observations included NP shape and total NP area; NP cell number and cellular morphology; appearance of the NP-AF border; lamellar organization and any tears or disruptions in the AF region; and disruptions, microfractures, osteophytes, or ossifications in the endplate. These observations were scored on a scale of 0 (non-degenerative) to 2 (severe degenerative changes), with total scores ranging from 0 (normal) to 16 (most severe). Two independent observers, blinded to the sample information, conducted the histological analysis of all samples.

Additionally, the acquired samples were dewaxed, rehydrated, and stained with hematoxylin and eosin (H&E) for the analysis of tissue morphology and the number of NP cells. The disc NP cell count and H&E-positive area were quantified using ImageJ.

### 4.14. Immunofluorescence and Immunohistochemistry

Immunohistochemical analysis was conducted for aggrecan and type II collagen, along with immunofluorescence analysis for calcitonin gene receptor protein (CGRP), Tie2, Brachyury, matrix metalloproteinase-13 (MMP-13), human nuclei antibody, tumor necrosis factor-alpha (TNF-α), and interleukin-1-beta (IL-1β). For immunostaining, the sections were dewaxed, rehydrated, and stained with primary antibodies against aggrecan (1:1000) (ab36861; Abcam, Cambridge, UK), type II collagen (1:100) (ab34712; Abcam), CGRP (1:200) (ab47027; Abcam), Tie2 (1:200) (NBP2-20636; Novus Biologicals, Littleton, CO, USA), Brachyury (1:200) (sc-166962; Santa Cruz Biotechnology), human nuclei antibody (1:200) (MAB1281; Sigma-Aldrich), TNF-α (1:200) (ab6671; Abcam), MMP-13 (1:200) (ab39012; Abcam), and IL-1β (1:200) (AF-501-NA; Novus Biologicals). After 24 h of incubation, the sections were rinsed with 1×PBS containing 0.1% Tween 20 and subsequently incubated with the secondary antibody anti-Rb horseradish peroxidase (Roche Diagnostics Ltd., Basel, Switzerland), as well as Alexa Fluor 488- (1:400) (A11034; Invitrogen), Alexa Fluor 488- (1:400) (A11029; Invitrogen), Alexa Fluor 568- (1:400) (A10042; Invitrogen), and Alexa Fluor 647 (1:400) (A21469; Invitrogen)-conjugated secondary antibodies. Following additional washes, the sections were counterstained with 4′,6-diamidino-2-phenylindole (DAPI) (1:1000) (Abcam) for 10 min. The prepared sections were then examined using a fluorescence microscope (Zeiss 880, Oberkochen, Germany and Leica SP5). The percentages of the area positive for aggrecan and type II collagen, as well as the number of positive cells relative to DAPI for CGRP, Tie2, Brachyury, MMP-13, human nuclei antibody, TNF-α, and IL-1β, were quantified using ImageJ.

### 4.15. Statistical Analysis

Statistical analysis was performed using GraphPad Prism (version 5.01; GraphPad Software, San Diego, CA, USA) and ImageJ. All data were represented as mean ± standard deviation. To determine statistical significance, one-way analysis of variance (ANOVA) with the Tukey post-test or two-way ANOVA was utilized. The threshold for statistical significance was established at *p* < 0.05.

## 5. Conclusions

In summary, we engineered ToMSCs with a tetracycline-off system (ToMSC-Tetoff-TGFβ1-IGF1-BMP7) as a potential treatment for IVD degeneration and compared its efficacy with other ToMSC implantations. Our findings reveal that the ToMSC-Tetoff-TGFβ1-IGF1-BMP7 treatment group displayed notably superior therapeutic effects compared with the injury-only, ToMSC-TGFβ1, and ToMSC-SDF1α implantation groups. The therapeutic benefits included: (1) stimulating the regeneration of NP cells, (2) restoring ECM structure by enhancing the expression of aggrecan and type II collagen, (3) downregulating pro-inflammatory cytokines, and (4) reducing the expression of neuropeptides in the nociceptive pathway, thereby manifesting both anti-inflammatory and pain suppressive effects. These outcomes suggest that ToMSCs equipped with the Tet-off system could potentially constitute an effective therapeutic strategy for IVD regeneration, inhibiting inflammatory factor secretion and suppressing pain in cases of human IVD degeneration.

## Figures and Tables

**Figure 1 ijms-24-16024-f001:**
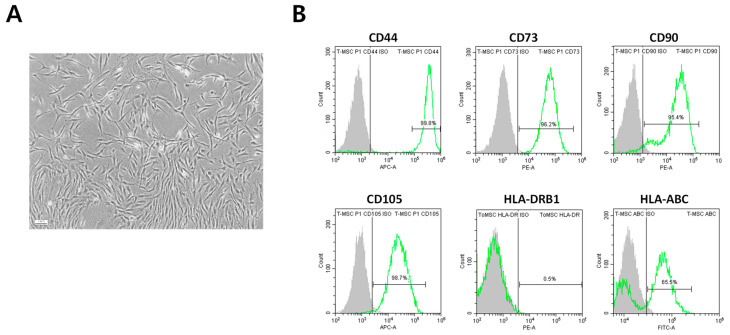
Morphology of tonsil-derived mesenchymal stem cells (ToMSCs) and flow cytometric analysis. (**A**) Morphology of ToMSCs under a light microscope (×40). Scale bar, 100 mm. (**B**) Flow cytometry analysis of ToMSCs using antibodies specific for typical MSC surface markers (CD44, CD73, CD90, and CD105) and HLA antigens (class I and II). (Grey: unstaining, Green: protein expression).

**Figure 2 ijms-24-16024-f002:**
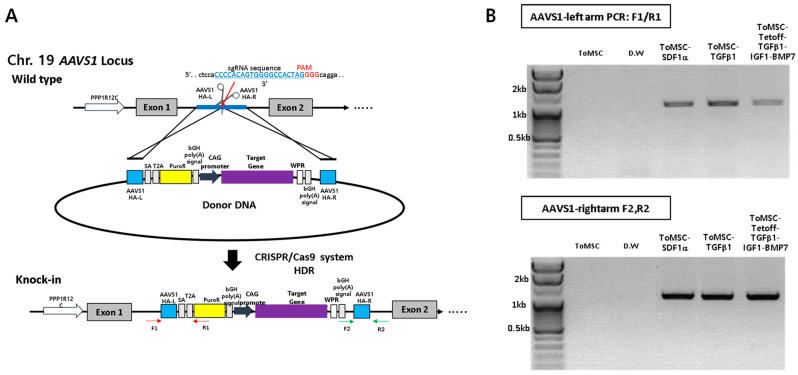
CRISPR/Cas9-mediated knock-in of transgenes into a safe-harbor site (AAVS1) on the ToMSC chromosome. (**A**) Schematic representation of the homology-directed repair-mediated knock-in process. (**B**) Confirmation of on-target knock-in by PCR (cropped image, uncropped gels image of Figure 2B is in Appendix A).

**Figure 3 ijms-24-16024-f003:**
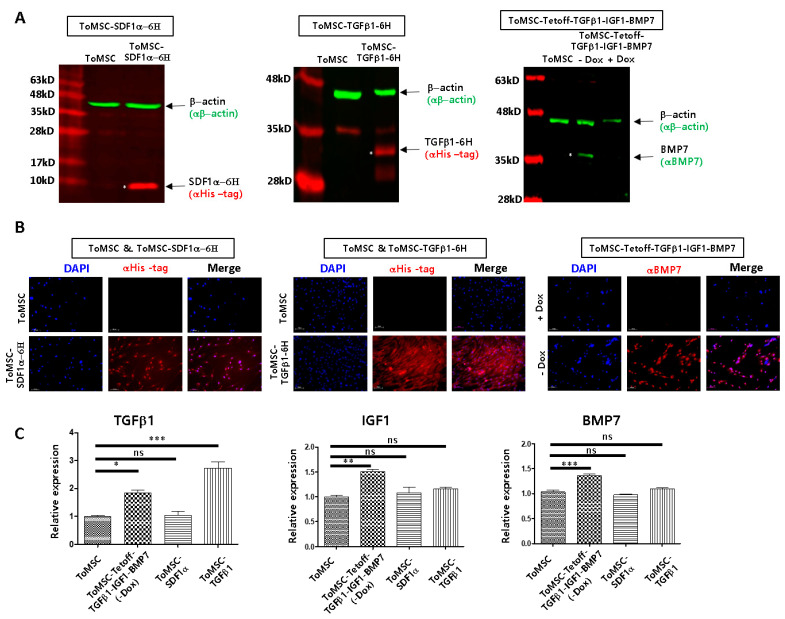
Transgenic expression as evaluated by western blots, immunocytochemistry, and qRT-PCR. (**A**) Western blot (cropped image, uncropped blots image of Figure 3A is in Appendix A), (**B**) immunostaining (100×) and (**C**) qRT-PCR were performed to examine the expression of transgenes. *** *p* < 0.001, ** *p* < 0.01, * *p* < 0.05, ns: not significant.

**Figure 4 ijms-24-16024-f004:**
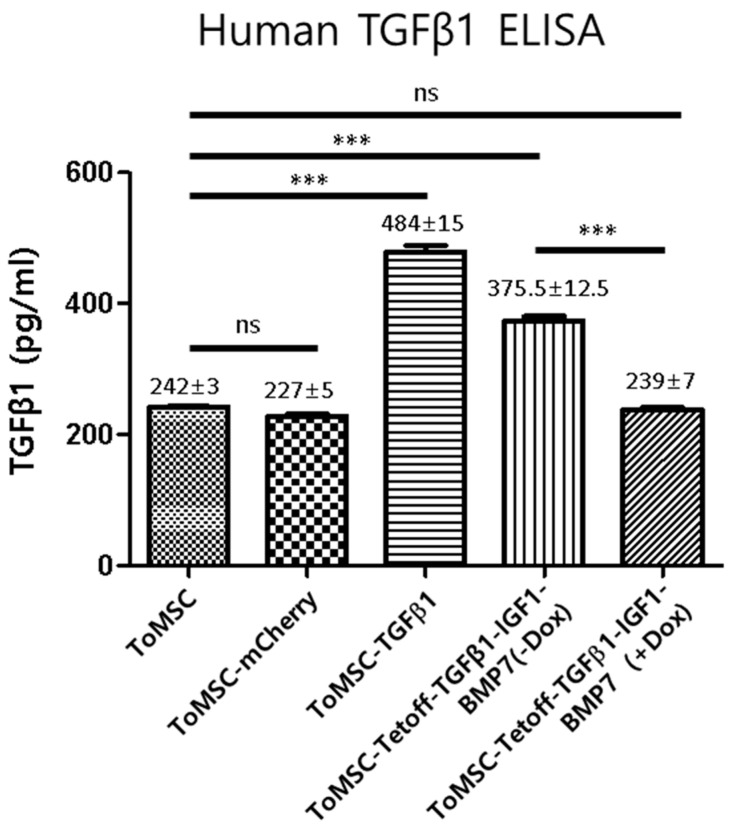
Expression of TGFβ1 as measured by ELISA. ELISA analysis of TGFβ1 expression in naïve and genetically engineered ToMSCs. *** *p* < 0.001. ns: not significant.

**Figure 5 ijms-24-16024-f005:**
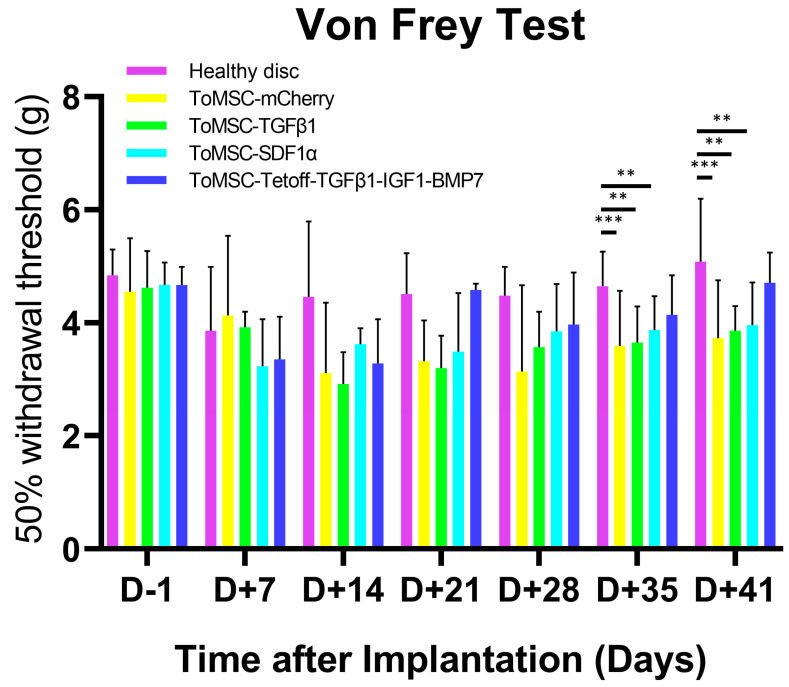
Reduction of the injury-induced pain phenotype by ToMSC-TetOff implantation. In the von Frey test, the 50% withdrawal threshold was significantly higher in the injury-only (G1), ToMSC-TGFβ1 (G2), and ToMSC-SDF1α (G3) implantation groups than in the ToMSC-Tetoff implantation group. ** *p* < 0.01, *** *p* < 0.001, indicating a significant difference between groups as determined by one-way analysis of variance.

**Figure 6 ijms-24-16024-f006:**
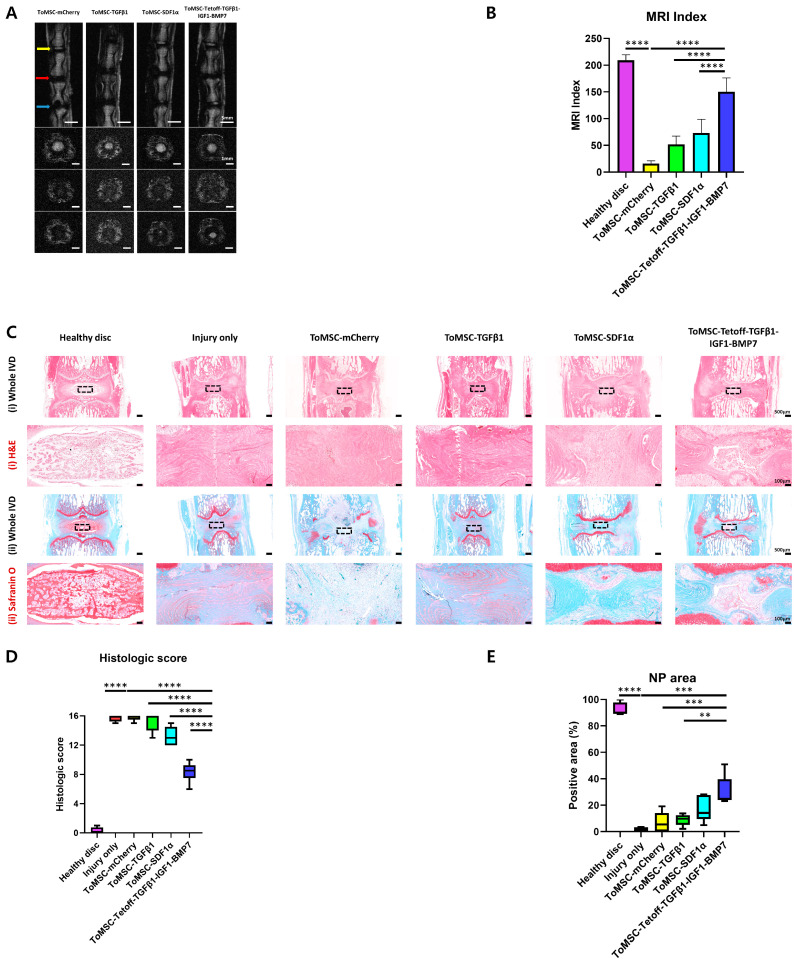
ToMSC-Tetoff injection significantly enhanced the repair of degenerated discs by restoring disc hydration and proteoglycan content in a rat needle puncture injury model. (**A**) T2-weighted MRI of rat coccygeal discs of the experimental groups taken 6 weeks after implantation. The yellow arrow indicates healthy discs (normal control). The red arrow indicates injury-only discs (injured control). The blue arrow indicates implanted discs. (**B**) Changes in the T2-weighted MRI index due to MSC implantation. (**C**) (i) Hematoxylin and eosin (H&E) staining results of discs from the needle puncture injury rat model at 6 weeks after treatment under low magnification (on **top**). Black rectangles indicate the area of the NP displayed at higher magnification (**bottom**). (ii) Safranin O staining results of discs from the needle puncture injury rat model at 6 weeks after treatment under low magnification (**top**). Black rectangles indicate the area of the NP displayed at higher magnification (**bottom**). (**D**) Changes in the histologic score based on H&E and Safranin O staining results. (**E**) Changes in the positive NP area ratio. ** *p* < 0.01, *** *p* < 0.001, **** *p* < 0.0001, indicating a significant difference between groups as determined by one-way analysis of variance.

**Figure 7 ijms-24-16024-f007:**
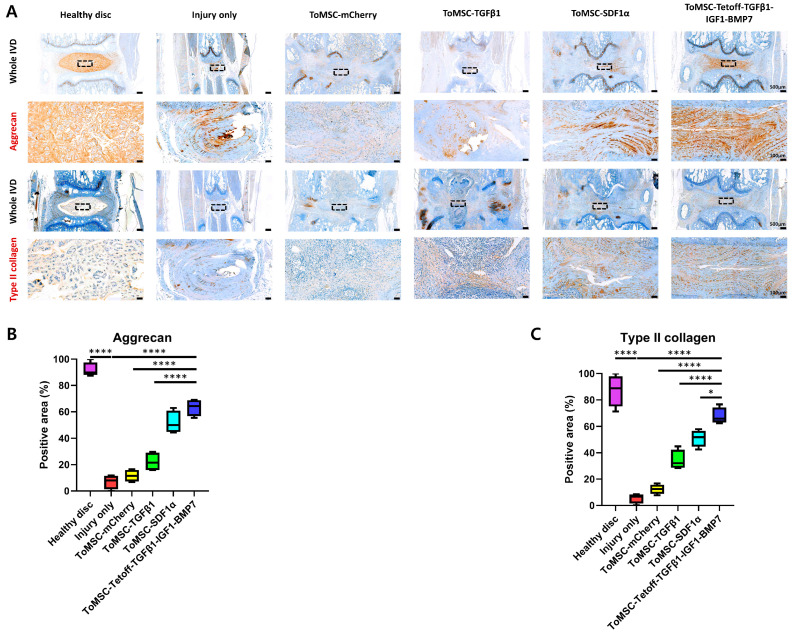
The ToMSC-Tetoff group demonstrated the greatest preservation of the matrix proteins in the disc nucleus pulposus (NP) of a rat needle puncture injury model. (**A**) Immunohistochemical staining of aggrecan (i) and type II collagen (ii) under low magnification (**top**) and higher magnification (**bottom**). Black rectangles indicate the area of the NP displayed at higher magnification (on **bottom**). (**B**) Changes the percentage of aggrecan-positive area in the disc NP spaces. (**C**) Changes in the percentage of type II collagen-positive area in the disc NP spaces. * *p* < 0.05, **** *p* < 0.0001, indicating a significant difference between groups as determined by one-way analysis of variance.

**Figure 8 ijms-24-16024-f008:**
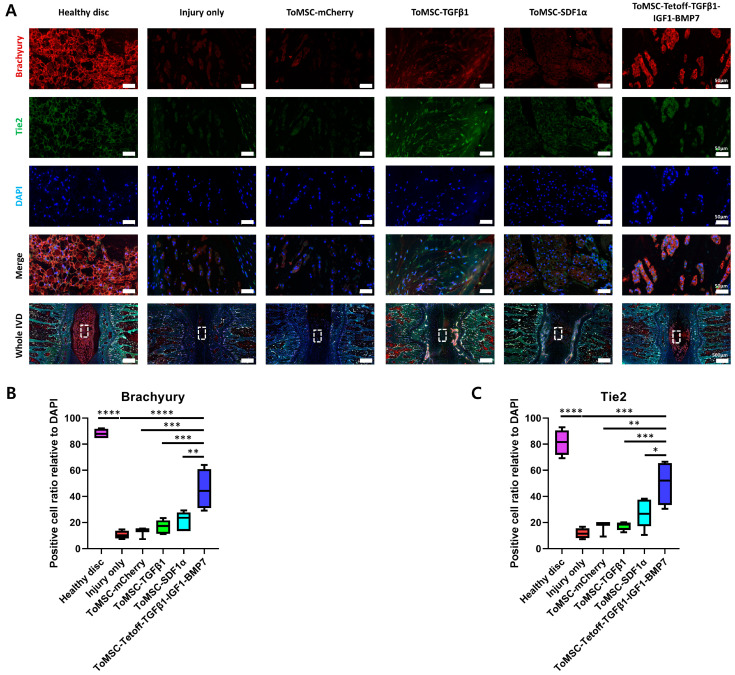
The ToMSC-Tetoff group demonstrated the best preservation of cells with the endogenous phenotype in disc nucleus pulposus (NP). (**A**) Immunofluorescence of Brachyury (red), Tie2 (green), DAPI (blue), and merged signals. White rectangles indicate the area of the NP displayed at higher magnification. (**B**) Changes in the percentage of Brachyury-positive cells in the disc NP space. (**C**) Changes in the percentage of Tie2-positive cells in the disc NP space. Immunopositivity was calculated in the disc NP using low-power fields and is expressed relative to the total number of DAPI-positive cells. * *p* < 0.05, ** *p* < 0.01, *** *p* < 0.001, **** *p* < 0.0001, indicating a significant difference between groups as determined by one-way analysis of variance.

**Figure 9 ijms-24-16024-f009:**
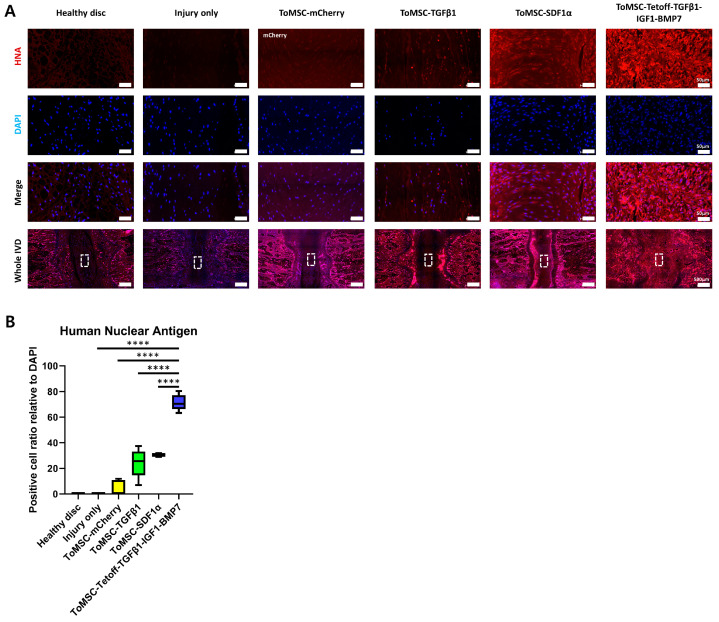
Retention of implanted stem cells in the disc nucleus pulposus of a rat tail needle puncture injury model at 6 weeks after cell implantation. (**A**) Immunofluorescence for human nuclear antigen (red), DAPI (blue), and merged signals at 6 weeks after treatment. White rectangles indicate the area of the nucleus pulposus displayed at higher magnification. (**B**) Changes in the percentage of human nuclear antigen-positive cells. Immunopositivity was calculated on disc nucleus pulposus with a low-power field and is expressed relative to the total number of DAPI-positive cells. **** *p* < 0.0001, indicating a significant difference between groups as determined by one-way analysis of variance.

**Figure 10 ijms-24-16024-f010:**
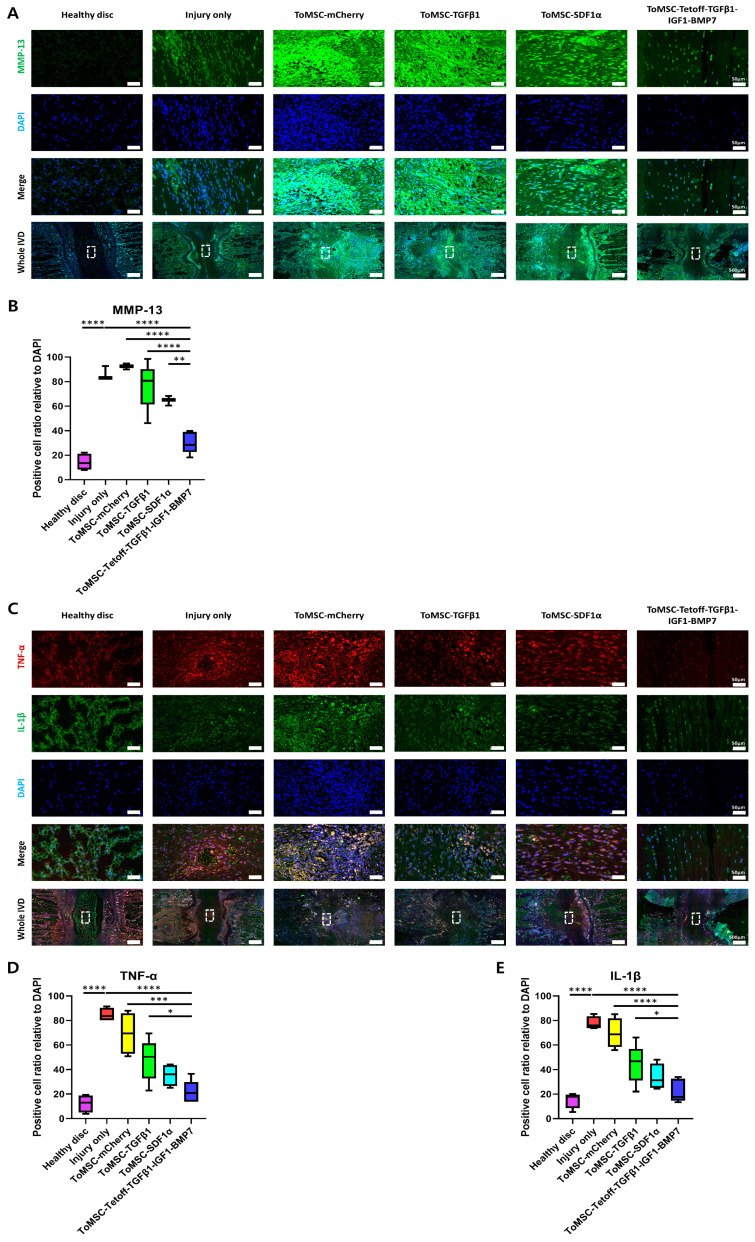
The ToMSC-Tetoff group demonstrated the greatest reduction in catabolic and pro-inflammatory cytokines in the disc nucleus pulposus (NP). (**A**) Immunofluorescence of rat tail discs for matrix metalloproteinase-13 (MMP-13) (green), DAPI (blue), and merged signals at 6 weeks after treatment. White rectangles indicate the area of the NP shown in higher-power fields. (**B**) Changes in the percentage of MMP-13-positive cells in the disc NP space. (**C**) Immunofluorescence of rat tail discs for tumor necrosis factor-alpha (TNF-α) (red) and interleukin (IL)-1β (green), DAPI (blue), and merged signals. White rectangles indicate the area of the NP displayed at higher magnification. (**D**) Changes in the percentage of TNF-α-positive cells in the disc NP space. (**E**) Changes in the percentage of IL-1β-positive cells in the disc NP space. Immunopositivity was quantified in the disc NP using low-power fields and expressed relative to the total number of DAPI-positive cells. * *p* < 0.05, ** *p* < 0.01; *** *p* < 0.001, **** *p* < 0.0001, indicating a significant difference between groups as determined by one-way analysis of variance.

**Figure 11 ijms-24-16024-f011:**
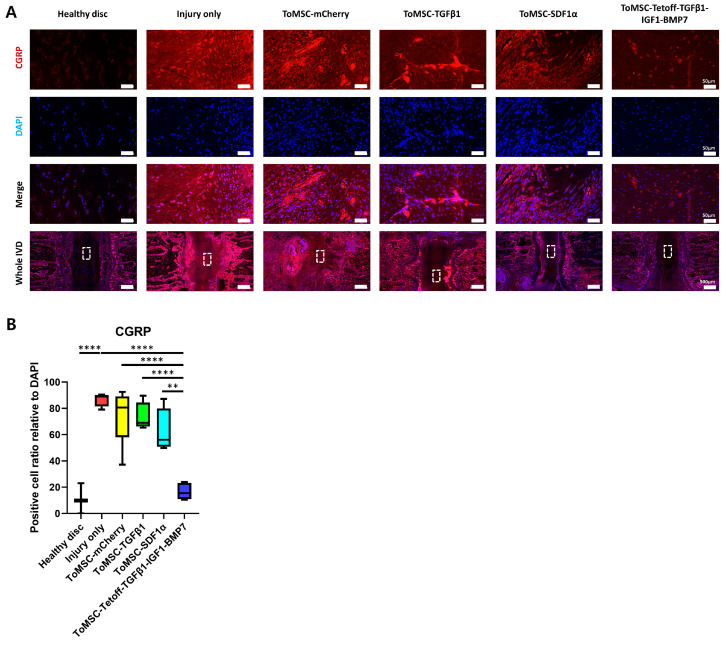
ToMSC-Tetoff treatment exhibits the strongest inhibition of injury-induced pain. (**A**) Immunofluorescence of rat tail discs for calcitonin gene receptor protein (CGRP) (red), DAPI (blue), and merged signals at 6 weeks after treatment. White rectangles indicate the area of the nucleus pulposus (NP) displayed at higher magnification. (**B**) Changes in the percentage of CGRP-positive cells within the disc NP spaces. Immunopositivity was quantified in the disc NP in low-power fields and expressed relative to the total number of DAPI-positive cells. ** *p* < 0.01, **** *p* < 0.0001, indicating a significant difference between groups as determined by one-way analysis of variance.

## Data Availability

The data that support the findings of this study are available from the corresponding author upon reasonable request.

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
