# Peer review of "Enhanced Intervertebral Disc Repair via Genetically Engineered Mesenchymal Stem Cells with Tetracycline Regulatory System"

_ijms, 2023, doi:10.3390/ijms242216024_

Round 1
Reviewer 1 Report
Comments and Suggestions for Authors
The manuscript proposes the use of MSC-TetOff-TGFβ1-IGF1-BMP7 cell transplant therapy for an intervertebral disc injury rat model. Overall, the paper is well written and organized. However, there are several serious drawbacks in the study design that impact the significance of the conclusion.
-
The use of human MSC cells xenograft in the rat model raises questions about the viability and maintenance of the injected human cells. Can the author provide evidence to support the viability and maintenance of these cells?
-
Figures 8-10 present the quantification of crucial factors through fluorescence images. However, the use of fluorescence images alone may not be convincing, especially considering that the contrast, brightness, and exposure time of each image may not be consistent (even the brightness of DAPI appears different). Therefore, additional methods such as Western blots, ELISA, qPCR, or flow cytometry should be employed to support the results.
-
Figure 5 is difficult to interpret and compare.
Well writen, minor editing of English language required.
Reviewer 2 Report
Comments and Suggestions for Authors
The manuscript titled "Enhanced Intervertebral Disc Repair via Genetically Engineered Mesenchymal Stem Cells with Tetracycline Regulatory System" is a comprehensive study that explores the therapeutic potential of genetically engineered mesenchymal stem cells (MSCs) in treating intervertebral disc (IVD) degeneration. Utilizing a tetracycline-off (Tet-off) system, the authors successfully engineered MSCs to express key factors—TGFβ1, IGF1, and BMP7—that are instrumental in disc tissue regeneration, inflammation suppression, and pain mitigation. The study demonstrates that these engineered MSCs remain viable post-implantation and effectively stimulate the proliferation of nucleus pulposus (NP) cells and the formation of extracellular matrix (ECM) structures. Additionally, the treatment led to a significant reduction in inflammation and pain-related markers like CGRP. Despite some limitations, such as the use of a rat model that may not fully replicate human IVD degeneration, the study is robust and offers a promising new avenue for treating IVD degeneration.
As for the quality of the manuscript, it appears to be well-structured and thorough, covering both the mechanistic aspects and the therapeutic implications of the engineered MSCs. It also acknowledges its limitations and suggests directions for future research, adding to its credibility. However, one area that could be improved is the quality of IHC images for better clarity and data interpretation. Overall, the manuscript makes a significant contribution to the field of regenerative medicine for IVD repair.
Below are some suggestions to enhance the quality of this manuscript:
Introduction
1. The introduction effectively sets the stage by discussing the prevalence and impact of IVD degeneration. However, it might be helpful to briefly mention the limitations of current treatments earlier to better set up the rationale for exploring MSCs.
2. Line 79 mentions various potential treatments like growth factors, gene therapy, etc. It would be beneficial to briefly explain why MSCs are particularly promising compared to these other avenues.
3. The discussion on MSCs is thorough but could be enhanced by citing more recent studies, if available, to strengthen the argument for their therapeutic potential.
4. Line 98 mentions MSCs can differentiate into chondrogenic cell types. It would be useful to elaborate on why this is particularly beneficial for treating IVD degeneration.
5. The explanation of the Tet-off system is detailed but might be too technical for readers not familiar with molecular biology. Consider simplifying this section or providing a brief summary for non-experts.
6. Lines 133-159 discuss the study objective and the specific MSCs used. This section would benefit from a clearer statement of the hypothesis being tested.
7. Line 136-139 talks about the advantages of Tonsil-derived MSCs (ToMSCs) over BMSCs and ADSCs. It would be helpful to cite studies that support these claims.
8. Lines 142-159 discuss the factors (TGF-β1, IGF-1, BMP-7) being modulated. It would be beneficial to explain why these specific factors were chosen and how they are expected to work synergistically.
9. Ensure that all the claims are backed by appropriate citations. For example, the statement in line 89 about the urgent need for research could be supported by a citation.
10. Consider adding a sentence or two discussing the limitations of using MSCs or the Tet-off system for treating IVD degeneration.
11. The text is mostly clear but could benefit from some minor grammatical corrections for readability. For example, in line 64, consider changing "processes" to "processes involved in IVD degeneration" for clarity.
Results
12. The characterization of ToMSCs is straightforward but could be enhanced by providing statistical data to back up the flow cytometry results.
13. The description of the CRISPR/Cas9 system is clear, but it would be helpful to mention why the AAVS1 locus was chosen as a safe-harbor site.
14. Consider elaborating on how the "precise insertion" was confirmed. Was it sequence-verified?
15. The methods used for confirming transgene expression are comprehensive. However, it would be beneficial to include the fold-change or other quantitative measures in the text, not just in the figures.
16. The use of "-Dox" and "in the absence of doxycycline" is a bit confusing. Consistency in terminology will improve clarity.
17. The time points for the von Frey test are well-described. However, the text could benefit from a brief explanation of why these specific time points were chosen.
18. Consider elaborating on the implications of the 50% withdrawal threshold findings. What does this mean in the context of IVD degeneration and pain relief?
19. The MRI and histological analyses are well-described. However, the text could be clearer on how these findings correlate with the functional outcomes like anti-allodynic effects.
20. The term "MRI index" is used but not defined. A brief explanation or citation would be helpful.
21. The text is mostly clear but could benefit from some minor grammatical corrections. For example, in line 276, consider changing "may play a role in delaying and repairing" to "may play a role in both delaying the progression of and repairing degenerated disc tissue" for added clarity.
22. The authors should clarify the sample size and statistical methods used for flow cytometric analysis. It would be helpful to include negative controls for the flow cytometry to strengthen the validity of the results.
23. The statistical methods used to analyze the von Frey test data should be clearly stated.
24. The authors should discuss the clinical relevance of their findings in the context of existing literature.
25. ToMSC-Tetoff-TGFβ1-IGF1-BMP7 demonstrated the greatest decrease in catabolic enzymes and pro-inflammatory cytokines:
a. The authors should elaborate on the implications of reduced MMP-13, TNF-α, and IL-1β expression.
b. It would be beneficial to discuss how these findings could translate to clinical applications.
26. The authors should discuss the limitations of using CGRP as a sole marker for pain and whether other markers were considered.
27. The quality of the IHC images could be improved for better clarity and interpretation. Poor-quality images can make it difficult for readers to validate the reported findings. High-resolution images with appropriate controls would add credibility and rigor to the study.
Discussion
28. The authors should consider discussing how their findings compare with other regenerative therapies for IVD degeneration, including stem cell-based therapies without the Tet-off system. While the study cites previous research on TGFβ1, IGF1, and BMP7, it could benefit from a more in-depth discussion of how these factors interact synergistically or additively in the context of ToMSC-Tetoff.
29. The authors acknowledge the limitations of the rat model but could further discuss how these limitations might affect the translatability of their findings to human subjects.
30. The authors should elaborate on potential clinical trials, including any plans for scaling up the technology for human use. It would be beneficial to discuss the long-term safety concerns associated with the Tet-off system, including the potential for uncontrolled cell proliferation or tumorigenesis.
31. The section is generally well-written but could be improved by breaking down some of the longer sentences for easier readability. The authors should consider summarizing the key findings and their implications in a concluding paragraph to provide a clear takeaway for the reader.
Comments on the Quality of English LanguageIf the manuscript maintains this level of language quality, it should meet the standards expected in international scientific journals. However, it's crucial to ensure that the manuscript is free from grammatical errors, awkward phrasing, or unclear terminology that could potentially confuse readers or reviewers.
Reviewer 3 Report
Comments and Suggestions for Authors
Authors examined the effects of human tonsil-derived MSC with inducible expression of TGFβ1, IGF1 and BMP7, all of which are known to accelerate the regeneration of intervertebral discs, by using a murine transplantation model. They have successfully demonstrated its therapeutic potential by showing an improvement of symptons and evidence for disc regeneration. This study will contribute to the development of regenerative medicine for the treatment of intervertebral disc diseases.
Author Response
We appreciate the reviewer's comments. We believe that these outcomes suggest that ToMSCs equipped with the Tet-off system could potentially constitute an effective therapeutic strategy for IVD regeneration, inhibiting inflammatory factor secretion and suppressing pain in cases of human IVD degeneration.
Round 2
Reviewer 1 Report
Comments and Suggestions for Authors
The response from authors addressed my concerns.
Reviewer 2 Report
Comments and Suggestions for Authors
Thank you to the authors for their thorough revisions. The quality of the manuscript has noticeably improved since the initial submission, and it's clear that the authors have made a commendable effort to address all the concerns I raised.
Comments on the Quality of English LanguageMinor editing of English language required